# Curiosity or Underdiagnosed? Injuries to Thoracolumbar Spine with Concomitant Trauma to Pancreas

**DOI:** 10.3390/jcm10040700

**Published:** 2021-02-11

**Authors:** Jakob Hax, Sascha Halvachizadeh, Kai Oliver Jensen, Till Berk, Henrik Teuber, Teresa Di Primio, Rolf Lefering, Hans-Christoph Pape, Kai Sprengel

**Affiliations:** 1Department of Trauma, University Hospital Zurich, 8091 Zurich, Switzerland; sascha.halvachizadeh@usz.ch (S.H.); KaiOliver.Jensen@usz.ch (K.O.J.); till.berk@usz.ch (T.B.); Henrik.teuber@gmail.com (H.T.); teresa.diprimio@usz.ch (T.D.P.); hans-christoph.pape@usz.ch (H.-C.P.); kai.sprengel@usz.ch (K.S.); 2Institute for Research in Operative Medicine (IFOM), Witten/Herdecke University, 51109 Cologne, Germany; rolf.lefering@uni-wh.de

**Keywords:** spine injury, thoracolumbar junction, pancreatic injury, trauma registry, acute care surgery, polytrauma management

## Abstract

The pancreas is at risk of damage as a consequence of thoracolumbar spine injury. However, there are no studies providing prevalence data to support this assumption. Data from European hospitals documented in the TraumaRegister DGU^®^ (TR-DGU) between 2008–2017 were analyzed to estimate the prevalence of this correlation and to determine the impact on clinical outcome. A total of 44,279 patients with significant thoracolumbar trauma, defined on Abbreviated Injury Scale (AIS) as ≥2, were included. Patients transferred to another hospital within 48 h were excluded to prevent double counting. A total of 135,567 patients without thoracolumbar injuries (AIS ≤ 1) were used as control group. Four-hundred patients with thoracolumbar trauma had a pancreatic injury. Pancreatic injuries were more common after thoracolumbar trauma (0.90% versus (vs.) 0.51%, odds ratio (OR) 1.78; 95% confidence intervals (CI), 1.57–2.01). Patients with pancreatic injuries were more likely to be male (68%) and had a higher mean Injury Severity Score (ISS) than those without (35.7 ± 16.0 vs. 23.8 ± 12.4). Mean length of stay (LOS) in intensive care unit (ICU) and hospital was longer with pancreatic injury. In-hospital mortality was 17.5% with and 9.7% without pancreatic injury, respectively. Although uncommon, concurrent pancreatic injury in the setting of thoracolumbar trauma can portend a much more serious injury.

## 1. Introduction

A flexion-distraction fracture is an unstable injury of the spine [1]. The proposed mechanism of injury is a flexion and distraction force on the spine over a fulcrum site [2,3,4], such as a seat belt in a car accident [3,4,5,6,7], but other common trauma mechanisms include falls [6,8,9,10]. In general, flexion-distraction injuries in adults are located at the thoracolumbar junction T11–L2 [8,11], and fractures of the thoracolumbar spine are common [8,9,12,13]. Due to its biomechanical properties, this region appears to be the weak point of the spine [1,6], representing the transition from the rigid rib cage to the most flexible lumbar spine [9,14,15], thereby making it more vulnerable to traumatic lesions [14]. Concomitant intra-abdominal injuries are often found in such fractures [3,5,8,16,17,18]. In the case of injuries due to seat belts, there is a direct compression of the abdomen, which results in compression of the intra-abdominal organs against the vertebral column [19,20]. The pancreas is a retroperitoneal organ that runs across the spine in the area of the upper lumbar vertebrae. Because of the anatomy of the pancreas, a linkage between these injuries during a high velocity trauma to the thoracolumbar spine can be expected [21], despite the pancreas being well protected because of its dorsal location in the abdomen. The retroperitoneal location is also responsible for the poor presence of clinical symptoms [20]; hence, a pancreatic injury is difficult to diagnose [19,22,23,24]. To make matters worse, pancreatic injuries are initially difficult to detect in computer tomography (CT) scan [25,26,27,28], which often leads to a delayed diagnosis. Moreover, increases in serum amylase and lipase laboratory parameters, if any, can only be observed a few hours after the injury, [27,29,30]. The typical clinical triad of upper abdominal pain, leukocytosis, and elevation of serum amylase levels is rarely pronounced [31]. However, early diagnosis of a traumatic pancreatic injury is very important because of possible complications and an increase of morbidity and mortality [25,32,33,34,35]. An undetected or delayed diagnosis of a traumatic pancreatic injury can cause severe retroperitoneal or abdominal inflammation, like pancreatitis [22,23,36], or in case of a disruption of the main pancreatic duct, a pseudopancreatic cyst [24,36]. Other complications include infection, abscess, duct stricture, and endocrine/exocrine insufficiency, which are associated with high morbidity and mortality [19,22,23,31,37]. This severely complicates the course of the disease for patients with pancreatic trauma with multiple injuries.

All literature regarding spinal injuries with concomitant pancreatic trauma refer to smaller case series. However, there is no description of the exact prevalence of this injury in a large trauma collective. Data from the TR-DGU were analyzed to estimate the prevalence of pancreatic trauma in thoracolumbar spine injuries and to determine its significance in clinical outcome.

## 2. Materials and Methods

### 2.1. TraumaRegister DGU^®^ (TR-DGU)

The TR-DGU of the German Trauma Society (Deutsche Gesellschaft für Unfallchirurgie, DGU) was founded in 1993. The purpose of this multi-center database is the pseudonymized and standardized documentation of severely injured patients.

Data are prospectively collected in four consecutive time phases from the site of the accident until discharge from hospital: (A) pre-hospital phase; (B) emergency room and initial surgery; (C) ICU, and (D) discharge. The documentation includes detailed information on demographics, injury pattern, comorbidities, pre- and in-hospital management, course of ICU, relevant laboratory findings including data about transfusion, and outcome of each individual. The inclusion criterion is being admitted to hospital via emergency room with a subsequent ICU care or reaching the hospital with vital signs and dying before admission to ICU.

The infrastructure for documentation, data management, and data analysis is provided by Academy for Trauma Surgery (AUC—Akademie der Unfallchirurgie GmbH, 80639 München, Germany), a company affiliated with the German Trauma Society. Scientific leadership is provided by the Committee on Emergency Medicine, Intensive Care and Trauma Management (Sektion NIS) of the German Trauma Society. Participating hospitals pseudonymize their data and submit them into a central database via a web-based application. Scientific data analysis is approved according to a peer review procedure laid down in the publication guideline of TR-DGU.

The participating hospitals are primarily located in Germany (90%), but a rising number of hospitals of other countries contribute their data as well (currently, these include Austria, Belgium, China, Finland, Luxembourg, Slovenia, Switzerland, The Netherlands and The United Arab Emirates). At present, ~33,000 cases from >650 hospitals are entered into the database annually.

Participation in TR-DGU is voluntary. However, for hospitals associated with TraumaNetzwerk DGU^®^, the entry of at least a basic data set is obligatory for quality assurance reasons.

This study has followed the publication guidelines of the TR-DGU and is registered as TR-DGU project ID 2019-014.

This study has been performed in accordance with the ethical standards laid down in the 1964 Declaration of Helsinki and its later amendments. Ethical approval was not needed due to retrospective use of medical records.

### 2.2. Inclusion and Exclusion Criteria

A retrospective study was performed that included patients documented between 2008 and 2017 in the TR-DGU by a European hospital. Only cases with blunt trauma and an ISS ≥ 9 were included. Patients who were transferred to another hospital within 48 h were excluded to prevent double counting. Patients with or without a minor (AIS ≤ 1) thoracolumbar spine injury served as the control group (flow chart, Figure 1). Thoracolumbar spine injury combines thoracic or lumbar injuries to the spine. In cases of injuries affecting both height segments, we analyzed the higher AIS value. All types of traumatic pancreatic injuries were included. Patients with all types of relevant thoracolumbar spine injuries with an AIS severity ≥ 2 were included in the study collective. For descriptive analysis, the frequency of pancreatic injuries in relation to the severity of the spinal column injury, as per the AIS, was performed.

### 2.3. Statistical Analysis

A descriptive data analysis was performed to investigate the prevalence of accompanying pancreatic trauma in thoracolumbar spine injuries. Categorical variables are presented with counts and percentages, and continuous values are shown as mean and standard deviation (SD). Due to the large number of patients included here, statistical testing was avoided since even minor and non-relevant differences would turn out to be formally significant. We rather calculated 95%CI for selected findings and OR. The statistical analysis was performed using Statistical Package for the Social Sciences (version 24, IBM Inc., Armonk, NY, USA).

## 3. Results

Total 44,279 out of 179,846 patients had a relevant thoracolumbar spine injury (AIS ≥ 2). The average age of the study collective was 43.1 ± 18.6 years in the group with pancreatic injury (*n* = 400) and 49.9 ± 20.2 years in the group without it (*n* = 43,879). Of the patients with pancreatic injuries, 68.0% were male. The most common causes of trauma in patients with a concomitant pancreatic injury were car (38.0%) or motorbike (17%) accidents and high falls (>3 m; 23.8%). Overall, 63.3% of the accidents with subsequent pancreatic injuries occurred in road traffic, in contrast to 48.5% in thoracolumbar spine injuries without involvement of the pancreas. The injury severity showed a mean ISS of 35.7 ± 16.0 in cases with a pancreatic injury and 23.8 ± 12.4 in cases without it. Most of the patients with a concomitant pancreatic injury had an ISS ≥ 25 (*n* = 287; 71.8%). Based on the severity of the spinal column injury, there were 317 cases (79.3%) in the group of pancreatic trauma patients with an AIS of 2 points of the thoracolumbar spine and only 51 cases (12.8%) with an AIS of 3 points. In 28% of cases, patients with thoracolumbar spine injuries presented a relevant (AIS > 3) accompanying injury of the head, regardless of the presence of a pancreatic injury. An accompanying limb injury was more common in patients with pancreatic injuries, in contrast to a thoracolumbar spine injury without pancreatic trauma (44.8% vs. 30.3%; Table 1).

The average hospital stay (32.8 vs. 22.1 days) and the length of stay (LOS) in an ICU (17.5 vs. 8.7 days) were longer in patients with a pancreatic injury. Moreover, complications such as sepsis (19.4% vs. 8.3%) and multi-organ failure (MOF; 54.0% vs. 25.7%) were more common in the group with a pancreatic injury. The in-hospital mortality was almost twice as high in patients with a pancreatic injury (17.5% vs. 9.7%; Table 2).

About three-quarters (*n* = 317; 79.3%) of patients with a concomitant pancreatic injury showed up with a contusion or minor laceration. The overall prevalence of pancreatic injury was 60.7 (0.61%; 95%CI, 0.57–0.64) per 10,000 patients in all cases (with and without a spine injury). In cases with a thoracolumbar spine injury, the prevalence was 0.90%. In cases without a spinal injury, the prevalence was 0.51%. Therefore, the OR for a pancreatic injury is 1.78 (95%CI, 1.57–2.01) in patients with a thoracolumbar spine injury. If cases with isolated head injuries were excluded from the control group, the prevalence rate would increase to 0.65% and OR would be 1.39 (95% CI, 1.23–1.58).

At last, it seems that moderate (AIS 2, OR 1.93; 95% CI 1.68–2.20) and very severe (AIS 5, OR 1.99; 95% CI 1.32–3.00) thoracolumbar spine injuries are more likely to be associated with an accompanying pancreatic injury in comparison with spinal injuries with an AIS of 3 (OR 1.16; 95% CI 0.87–1.54; Table 3).

## 4. Discussion

To our knowledge, this is the first study that has investigated the correlation between concomitant pancreatic injuries and the severity of trauma to the thoracolumbar spine in a large trauma collective. It can be concluded that concomitant pancreatic injuries are rare, with a prevalence of 0.90% in a group of severely injured patients with trauma of the thoracolumbar spine.

In our collective, middle-aged men in particular presented with an injury to the thoracolumbar spine, which is in line with the literature; however, the men in literature were younger on average [8,9]. The mean age in our collective is also higher in the group with concomitant pancreatic injury than in the literature [32]. The trauma mechanisms that are primarily presented in our analysis coincide with the literature [5,6,8,9,32].

In our collective, the general occurrence of a traumatic pancreatic injury in the patients with multiple traumas turned out to be rare with a ratio of 6/1000 patients, which is in line with the literature [19,22,38]. However, patients with moderate to severe spinal injuries more often presented an accompanying pancreatic trauma (OR 1.78). It was shown that these concomitant pancreatic injuries occur primarily in spine injuries with a severity of AIS 2 (*n* = 317; OR 1.93). In our collective, AIS 2 in spinal injuries includes vertebral body fractures with minor compression (≥20% loss of anterior height), undislocated fractures of facet joints or nerve root injuries. However, AIS 2 excludes a fracture of the vertebral body with major compression (>20% loss of anterior height; AIS 3).

There was a decrease in pancreatic trauma cases combined with thoracolumbar spine injuries with a severity of AIS ≥ 3. Although the OR for the occurrence of a pancreatic injury increased in patients with severe thoracolumbar spine injuries (AIS 4, OR 1.87; AIS 5, OR 1.99), note that the number of cases was significantly lower (AIS 4, *n* = 8; AIS 5, *n* = 24). Because of the anatomical conditions of the pancreas in the retroperitoneum, a high velocity trauma of the abdomen is necessary to cause pancreatic trauma [22]. This is highlighted by the fact that patients with pancreatic trauma showed up with more severe injury patterns (mean ISS = 35.7) in comparison with thoracolumbar spine injuries without pancreatic trauma (mean ISS = 23.8). Likewise, in the literature, patients with traumatic pancreatic injuries were more seriously injured on average [32]. In some cases, the main force may be absorbed by the abdomen. Therefore, the spinal injury is less severe in comparison with the abdominal; this could be a potential explanation for the decreased probability of concomitant pancreatic trauma between the injury severity groups AIS 2 and AIS 3 (OR 1.93 vs. 1.16) in thoracolumbar spine injuries. This should be examined in future biomechanical studies. The pancreas appears to be at risk of injury because of its location across the thoracolumbar spine and its fixed retroperitoneal location.

Based on our observations, there is no direct correlation between the severity of a spinal injury and the occurrence of a concomitant pancreatic trauma. Nevertheless, as aforementioned, there was an accumulation of pancreatic injuries in moderate thoracolumbar spine injuries. Therefore, knowledge of the epidemiology of thoracolumbar spine injuries and accompanying trauma of surrounding organs is useful and important for the assessment, decision-making, and treatment of patients who have suffered a blunt trauma and are admitted to the emergency room.

Both a pancreatic injury [19,21,31,39] and a thoracolumbar spine injury can be clinically overlooked. Because of the often delayed or missing diagnosis of a pancreatic injury [24,25,32], the reported prevalence may be underestimated in our study and in the literature. Therefore, special attention should be paid to the spine and pancreas in the initial diagnostic investigation of a patient who has suffered a blunt abdominal trauma with hyperflexion. A missing diagnosis of a pancreatic injury can lead to complications in the course of treatment [19,22,31,37,38,40] and long-term consequences like diabetes [36]. This fact was also evident in our collective with increased complications such as sepsis and MOF in patients with a concomitant pancreatic injury. The presence of pancreatic trauma is associated with high rates of morbidity and mortality [22,32,38,41,42]. In our collective, the in-hospital mortality with accompanying pancreatic injury was also increased, as well as the LOS in the ICU and the length of time in hospital in general. This is because of the increased occurrence of complications and it illustrates the clinical relevance of the simultaneous occurrence of a pancreatic injury. In this context, a pancreatic injury does not represent an independent risk factor. The effect of a pancreatic injury (adjusted OR 0.88; 95% CI 0.63–1.25) is mostly represented by the Revised Injury Severity Classification II via the AIS of the most severe and second most severe injuries.

Diagnostically, a CT scan [21,23,26,28] and laboratory testing [29,30] of pancreatic amylase and lipase are important diagnostic tools; however, both can initially be negative [23,25,27,29]. Therefore, although concomitant pancreatic injuries might be a curiosity, the underdiagnosis of these concomitant injuries should be minimized, especially in thoracolumbar spine injuries with a severity of AIS 2. How many of these cases are effectively overlooked in daily clinical practice should be evaluated in further clinical studies.

### Study Limitations

Due to register evaluation, this study has some limitations. It contains only retrospective data that were registered in the TR-DGU. If a pancreatic injury was not diagnosed, it was not recorded in the register. Therefore, the overall prevalence of a pancreatic trauma can be underestimated. The level of injury bases on the highest AIS of either the thoracic or the lumbar spine. AIS was developed to classify injury severity. More sophisticated classifications of spinal injuries might provide more information about the nature of the injury. Unfortunately, the utilized database does not include the specific spinal classification. In the AIS version of 2005 (update in 2008), ligamentous injuries are not explicitly recorded. This limitation was corrected with the update of the AIS code in 2015; these injuries are coded as AIS 3 [43]. Based on the anonymized structure of the database, and the multicentric data entry, recoding is not possible. In order to increase comparability amongst included hospitals in this multicentric, international study, we utilized the AIS scoring system. We are aware of this limitation; however, we feel that the AIS is an adequate estimate of the injury severity. There is a higher rate of patients with ISS ≥ 25 in the pancreatic injury group (Table 1). However, the ISS cannot distinguish between direct injury of the intraabdominal organs or spinal injury. Therefore, we based this analysis on AIS spine.

The association of a thoracolumbar spine injury with concomitant trauma to the pancreas might depend on the specific type of seat belt (2-point vs. 3-point). This specific variable is not included in the registry. However, we believe that most patients had a 3-point seat belt as we included patients from 2008 to 2017 from European hospitals.

Epidemiological data, in particular, were presented in this study. The clinical relevance—as well as the influence on the outcome—has to be evaluated in future clinical prospective studies.

## 5. Conclusions

Concomitant pancreatic trauma in the case of thoracolumbar spine injuries is rare. However, a minor correlation could be seen between patients with a moderate spinal column injury (AIS 2) and patients with mild (AIS ≤ 1) injuries (OR 1.93). Therefore, in cases of moderate thoracolumbar spine injuries, a concomitant injury of the pancreas should be considered in order not to overlook the injury.

## Figures and Tables

**Figure 1 jcm-10-00700-f001:**
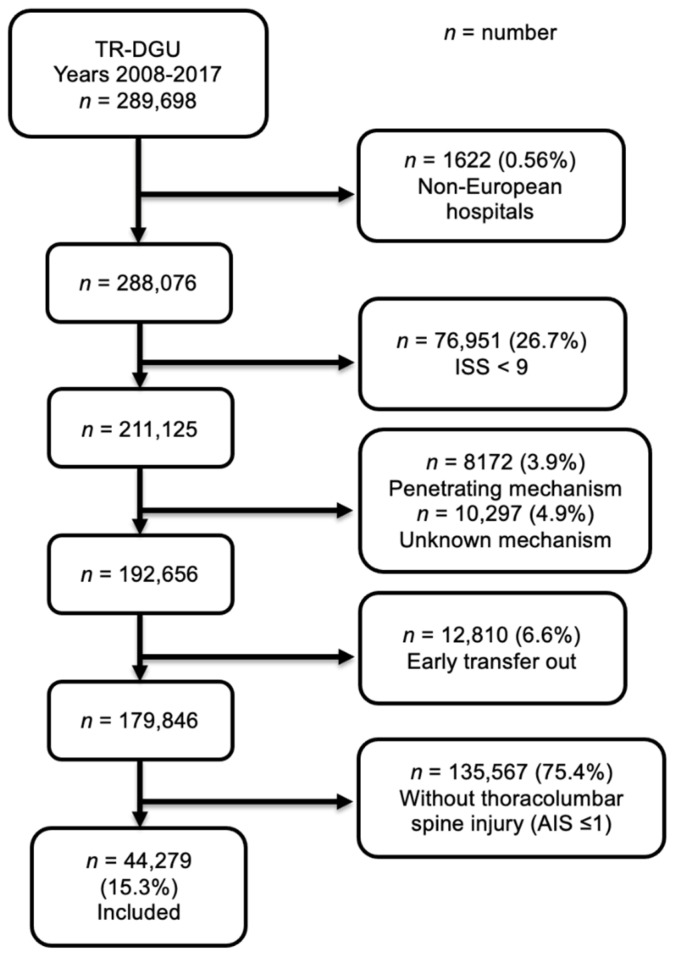
Patient inclusion flowchart.

**Table 1 jcm-10-00700-t001:** Demographic- and trauma-related data.

	Without Pancreatic Injury *n* = 43,879	With Pancreatic Injury *n* = 400
Age [years], mean (SD)	49.9 (20.2)	43.1 (18.6)
Male gender, *n* (%)	31,505 (71.8)	272 (68.0)
Severity of thoracolumbar trauma		
AIS 2, *n* (%)	32,103 (73.2)	317 (79.3)
AIS 3, *n* (%)	8597 (19.6)	51 (12.8)
AIS 4, *n* (%)	832 (1.9)	8 (2.0)
AIS 5, *n* (%)	2347 (5.3)	24 (6.0)
ISS [points], mean (SD)	23.8 (12.4)	35.7 (16.0)
ISS ≥ 25 points, *n* (%)	16,525 (37.7%)	287 (71.8)
Injury mechanism		
Traffic—car, *n* (%)	10,443 (23.8)	152 (38.0)
Traffic—motorbike, *n* (%)	5924 (13.5)	68 (17.0)
Traffic—bicycle, *n* (%)	2545 (5.8)	14 (3.5)
Traffic—pedestrian, *n* (%)	2369 (5.4)	19 (4.8)
High fall (>3 m), *n* (%)	13,295 (30.3)	95 (23.8)
Low fall (<3 m), *n* (%)	6231 (14.2)	15 (3.8)
Relevant head injury, AIS > 3, *n* (%)	12,549 (28.6)	114 (28.5)
Relevant thoracic trauma, AIS > 3, *n* (%)	24,616 (56.1)	323 (80.8)
Injuries of the extremities, AIS ≥ 2, *n* (%)	13,295 (30.3)	179 (44.8)

AIS = Abbreviated Injury Scale; SD = Standard Deviation; ISS = Injury Severity Score.

**Table 2 jcm-10-00700-t002:** Outcome parameters.

	Without Pancreatic Injury *n* = 43,879	With Pancreatic Injury *n* = 400
LOS [days], mean (SD)	22.1 (21.7)	32.8 (30.2)
ICU LOS [days], mean (SD)	8.7 (12.4)	17.5 (18.4)
In-hospital mortality, *n* (%)	4256 (9.7)	70 (17.5)
Sepsis *, *n* (%)	1878 (8.3)	40 (19.4)
MOF *, *n* (%)	5949 (25.7)	114 (54.0)

LOS = Length of Stay; ICU = Intensive Care Unit; MOF = Multi-Organ Failure. * Available only for patients with standard documentation.

**Table 3 jcm-10-00700-t003:** Risk of injured pancreas depending on severity of thoracolumbar injury.

AIS Thoracolumbar	OR	CI 95%
≤1	(reference)	
2	1.93	1.68–2.20
3	1.16	0.87–1.54
4	1.87	0.93–3.78
5	1.99	1.32–3.00
2–5	1.78	1.57–2.01

AIS = Abbreviated Injury Scale; OR = Odds Ratio; CI = Confidence Interval.

## Data Availability

The data that support the findings of this study are available at AUC GmbH, TR-DGU; however, restrictions apply to the availability of these data, which were used under license for the current study, and so are not publicly available. Data are, however, available from the authors upon reasonable request and with permission of AUC GmbH, TR-DGU.

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
