# Peer review of "Curiosity or Underdiagnosed? Injuries to Thoracolumbar Spine with Concomitant Trauma to Pancreas"

_jcm, 2021, doi:10.3390/jcm10040700_

Round 1
Reviewer 1 Report
This paper is a well written retrospective registry study on the coincidence of thoracolumbar spine injuries and pancreatic injuries and its potential impact on clinical outcome.
The study was performed using the trauma registry DGU® between 2008 – 2017 on 179,846 patients (44,279 patients with spine fractures and 135,567 patients without)
The authors revealed that pancreatic injuries were more common with spinal fractures. The majority was male, and there was an increased injury severity score (ISS).
This paper is an important contribution to the knowledge on this rare injury of the pancreas. However, in my opinion, there are some issues to be addressed to further improve the quality of this paper.
My specific comments:
Introduction:
The authors correctly describe that flexion-distraction injuries of the thoraco-lumbar junction are considered to be associated with pancreatic injuries.
Line 42: The association with seatbelts may be predominantly seen in specific ancient seatbelts, just fixing the pelvis/abdomen. The authors are encouraged to clarify if, this mechanism is also seen with modern 3 point seat belts in the cited literature.
Methods: How was a thoraco-lumbar injury found by using the AIS Coding? To my knowledge there are AIS codes for injuries to the thoracic- as well as the lumbar spine. The authors should clarify.
Results:
Some Results are presented with comparative statistics. This should be included/explained in the methods section.
Discussion:
Line 184: I assume that the used wording has been taken from the AIS Coding text. If so, the authors may considerer rephrasing because a “minor vertebral fracture” in the AIS can still be a severe burst or flexion/distraction injury.
The authors should avoid abbreviations (like MOF, LOS, ICU…) without explanation. Please considerer to write the full name the first time in the text.
The authors should present the limitation of this interesting and important study a little bit more in detail:
Definition of level of injury
Definition of fracture type – This is important because the introduction includes an association with flexion-distraction injuries which – to my knowledge- cannot be classified with the AIS Coding.
Reviewer 2 Report
This manuscript examines the prevalence of pancreatic trauma in blunt trauma patients with thoracolumbal spine injuries. The authors evaluated patient data documented in the TraumaRegister DGU between 2008 and 2017 related to the injury pattern, injury mechanism, severity and outcome.
This short performed analysis presents the risk to be considered of concomitant injured pancreas depending of moderate thoracolumbar injury but there are still a few minor aspects to comment.
Comments:
- The title with „Curiosity or Underdiagnosed? ...“ presents an interesting question that cannot be answered with the applied methods and remains open.
- In the abstract, p-values are given that were not described in material and methods or were not used in the results.
- Throughout the article, thoracolumbar spine injuries are mentioned. How exactly was this defined?
- Depending on the location of the spinal injury, an assignment to AIS thorax or abdomen is made for the calculation of the ISS and thus results in differences in the score calculation. Are there any differences in the incidence of pancreatic injury or outcome due to this fact? Furthermore, there was no correlation of pancreatic injury with outcome.
- According to the inclusion and exclusion criteria, patient data from European hospitals were used. According to Figure 1, patients who were not entered in German hospitals were excluded. What is the explanation?
- The tables, especially Table 1, are very confusing: There are formatting differences in the blank lines. Inconsistent naming of SD behind the Mean values or in a new row. Which AIS value was used for the extremity injuries? Inconsistent use of absolute values or percentages.
- The description of the results jumps between the different tables, which should be ordered.
- Check for consistency in spelling, e.g. line 148: use „AIS 3+“ otherwise ≥ is always used.
- Line 152/153: The description „contusion or minor laceration“ refers to the pancreatic injury and not the AIS value of 2 of the spinal injury as the sentence suggests.
